

# TanSat-2: a new satellite for mapping solar-induced chlorophyll fluorescence at both red and far-red bands with high spatio-temporal resolution

Dianrun Zhao[1,2,3], Shanshan Du[1,2], Chu Zou[1,2,3], Longfei Tian[4], Meng Fan[1,5], Yulu Du[1,2,3], and Liangyun Liu[1,2,3]

[1]Key Laboratory of Digital Earth Science, Aerospace Information Research Institute, Chinese Academy of Sciences, Beijing 100094, China;

[2]International Research Center of Big Data for Sustainable Development Goals, Beijing 100094, China

[3]University of Chinese Academy of Sciences, Beijing 100049, China

[4]Innovation Academy for Microsatellites of CAS, Shanghai 201203, China

[5]State Key Laboratory of Remote Sensing Science, Aerospace Information Research Institute, Chinese Academy of Sciences, Beijing 100101, China

*Correspondence to*: Shanshan Du (duss@radi.ac.cn)

**Abstract.** Global observations of solar-induced chlorophyll fluorescence (SIF) serve as a robust proxy for monitoring vegetation photosynthetic activity and elucidating the terrestrial carbon cycle. To date, several atmospheric remote sensing satellites have been deployed to generate global SIF products. However, accurate mapping of dual-band (red and far-red) SIF with daily temporal resolution and kilometer-level spatial resolution remains a critical gap, despite its significance for various applications. The Chinese next-generation greenhouse gas monitoring satellite, TanSat-2, is set to succeed the original TanSat satellite, aiming to record the fraction of greenhouse gases, pollutants, and SIF measurements from space. According to current schedules, TanSat-2 is slated for launch in 2026. This satellite will feature a wide swath of 2900 km, high spatial resolution of 2 km at an orbit altitude of 7000 km, and near-daily global coverage. TanSat-2 is equipped with two spectral channels—747–777 nm and 672–702 nm—operating at a spectral resolution of 0.12 nm, thereby offering significant potential for mapping SIF in both the red and far-red bands. In this study, we explore the prospective capabilities of TanSat-2 for SIF retrieval through simulation experiments. First, we simulated the satellite's radiative transfer processes using the Moderate Resolution Atmospheric Transmission 5 (MODTRAN 5) and Soil Canopy Observation of Photosynthesis and Energy (SCOPE) models. An end-to-end orbit simulation dataset for TanSat-2 was generated by aggregating global bottom-of-atmosphere (BOA) reflectance, meteorological datasets, and the global OCO-2 SIF dataset (GOSIF). We then assessed the theoretical accuracy of TanSat-2 based on this spectral simulation dataset, yielding root mean square error (RMSE) values of 0.24 and 0.19 mW m$^{-2}$ sr$^{-1}$ nm$^{-1}$ for SIF retrievals at 740 nm and 685 nm, respectively. Finally, we examined the global prospects of TanSat-2 SIF retrievals using the end-to-end orbit simulations. Comparisons between the anticipated TanSat-2 SIF retrievals and GOSIF inputs revealed excellent correlation at both bands, with R² values of 0.88 and 0.61, and RMSE values of 0.082 and 0.061 mW m$^{-2}$ sr$^{-1}$ nm$^{-1}$, respectively. Thus, TanSat-2 is poised to provide a valuable data resource for reliable SIF retrievals in the red and far-red bands, characterized by high spatio-temporal resolution on a global scale.



## 1 Introduction

Solar-induced chlorophyll fluorescence (SIF) is a subtle emission produced by chlorophyll during photosynthesis, spanning approximately 640–850 nm with distinct peaks around 685 and 740 nm (Mohammed et al., 2019; Guanter et al., 2014;). Recent advancements in hyperspectral satellite sensor technology have made global SIF monitoring a focal point of research (Guanter et al., 2012; Köhler et al., 2015b). SIF mapping from satellites offers a robust tool for assessing global vegetation photosynthetic activity and understanding the terrestrial carbon cycle (Zheng et al., 2024).

Table 1 outlines the specifications of satellites and instruments capable of SIF retrieval. The first global SIF map utilized data from the Japanese Greenhouse Gases Observing SATellite (GOSAT), which launched in 2009. GOSAT operates within a spectral range of 754-773 nm and boasts a spectral resolution of 0.025 nm. Its coverage includes a swath width of 790 km and a spatial resolution of 10.5 km diameter, providing moderate-resolution surface data but lacking in continuous global coverage due to periodic sampling intervals (Frankenberg et al., 2011b; Joiner et al., 2011). Conversely, the Orbiting Carbon Observatory-2 (OCO-2) enhances SIF retrieval capabilities with a 100-fold increase in observations compared to GOSAT, although it has a narrower swath of 10.3 km and a pixel size of 1.3 km × 2.25 km, still falling short of continuous global coverage (Frankenberg et al., 2014a; Sun et al., 2017). The subsequent OCO-3, launched in 2019, maintains similar specifications with a swath width around 10 km and a spatial resolution of 1.6 × 2.2 km (Taylor et al., 2020). Launched in 2016, the Chinese Carbon Dioxide Observation Satellite Mission (TanSat) covers a spectral range of 758-778 nm and offers a spectral resolution of 0.044 nm, with a swath width of 20 km and a spatial resolution of 2 km, facilitating SIF retrieval at Fe (centered at 758.8 nm) and KI (centered at 771 nm) Fraunhofer lines (Du et al., 2018). The recently launched First Terrestrial Ecosystem Carbon Inventory Satellite (TECIS-1) in August 2022, operates across 670-780 nm with a swath width of 34 km and an unprecedented spatial resolution of 375 m, suitable for dual-band (red and far-red) SIF retrieval (Du et al., 2020; Zou et al., 2021; Du et al., 2022). The Global Ozone Monitoring Experiment 2 (GOME-2) on the MetOp-A/B satellites, launched by Europe in 2007, operates in the red and near-infrared spectra (Joiner et al., 2013, 2016; Köhler et al., 2015). Initially featuring a pixel resolution of 40 km × 80 km, this was later refined to 40 km × 40 km for MetOp-A starting July 2013. GOME-2 scans the Earth with a 1920 km wide swath, achieving global coverage approximately every 1.5 days. Despite its relatively coarse resolution, GOME-2, along with the Scanning Imaging Absorption Spectrometer for Atmospheric Chartography (SCIAMACHY) aboard ENVISAT, has facilitated the creation of two global, spatially continuous solar-induced fluorescence (SIF) datasets. SCIAMACHY covers a spectral range of 595–812 nm at 0.48 nm resolution, extending across a 2800 km swath with pixel sizes varying from 30 km × 240 km to 30 km × 60 km (Joiner et al., 2012; Köhler et al., 2015a).

Launched in 2017, the TROPOspheric Monitoring Instrument (TROPOMI) on the Sentinel-5 Precursor offers unprecedented detail with its 2600 km swath and high spatial resolution of 3.5 km × 5.5/7.5 km, ensuring daily global coverage (Guanter et al., 2011; Köhler et al., 2018; Zhao et al., 2022a). As the first imaging spectrometer of its kind, TROPOMI provides kilometer-level spatial resolution, continuous spectral sampling from 675 to 775 nm at a resolution of 0.38 nm. While it delivers enhanced radiometric sensitivity and broad spectral coverage, its spectral resolution remains suboptimal for detailed red SIF retrieval (Zou et al., 2022).

Looking ahead to post-2025, the planned FLEX mission aims to survey the 500–780 nm range with an adaptable spectral resolution between 0.3 nm and 2.0 nm. It will feature a swath width of at least 150 km and a spatial resolution as fine as 0.3 km (Vicent et al., 2016; Coppo et al., 2017). This mission is set to significantly advance our monitoring of photosynthesis and vegetation health. Concurrently, the CO2M mission, slated for a 2026 launch, will offer high spectral and spatial resolutions. However, its narrower swath will limit its coverage area, impacting its ability to monitor SIF extensively.

The next-generation Chinese greenhouse gas monitoring satellite, TanSat-2, is scheduled for launch in 2026 as the successor to the original TanSat satellite. Unlike its predecessor, which primarily focused on carbon dioxide monitoring, TanSat-2 is designed





to facilitate global carbon stocktaking. To achieve this objective, it will combine a wide swath of 2,900 km with high spatial
resolution (2 km at an orbit altitude of 7000 km) and near-daily global coverage. Compared to the TROPOMI satellite, TanSat-2's
broader swath and enhanced spatial resolution are expected to yield more frequent and detailed observations, thereby improving
the quality and utility of data collected for environmental monitoring and research. Additionally, TanSat-2 will record upwelling
radiances with a spectral resolution of 0.12 nm across both the $O_2$-A and $O_2$-B bands. This extensive spectral coverage and high
spectral resolution will allow for a more accurate exploitation of the entire SIF spectrum, rather than being limited to the far-red
SIF.

8        Red SIF is recognized for its superior ability to reflect the biochemical characteristics and photosynthetic capacity of vegetation

compared to far-red SIF (Verrelst et al., 2015). However, the absorption features in the red region tend to be narrower and less
pronounced than those in the far-red band, complicating the retrieval of SIF in the red spectral region. Furthermore, there are
currently few satellites equipped with red SIF bands, resulting in ongoing challenges with poor signal quality and insufficient
accuracy at the application level for existing red SIF products (Dechant et al., 2022). Consequently, most satellite-based SIF
products have focused on the far-red band. Some studies have suggested that a payload with high spectral resolution (0.1 nm) and
a high signal-to-noise ratio (SNR > 1500) could fulfill the requirements for retrieving red SIF, as demonstrated through simulations
of satellite remote sensing accuracy (Zou et al., 2022). Presently, existing spaceborne imaging spectrometers, including TROPOMI,
TECIS-1, GOME-2, and FLEX, do not meet the stringent spectral requirements necessary for effective red SIF retrieval. Therefore,
TanSat-2 is poised to be the only satellite capable of delivering high-quality SIF retrievals for both the red and far-red bands (Zou
et al., 2022). With its wide swath of 2,900 km and spatial resolution of 2 km, TanSat-2 is anticipated to significantly enhance global
SIF observation capabilities compared to all current satellites.

20       In this study, we aim to (1) generate an end-to-end orbit simulation dataset for the TanSat-2 satellite; (2) optimize the empirical

parameters of the data-driven algorithm for red and far-red SIF retrieval; (3) investigate TanSat-2's potential for improving SIF
retrieval accuracy; and (4) explore the prospects of TanSat-2 for global-scale SIF monitoring.
**Table 1.** Specifications of the launched or scheduled satellites with the potential of SIF retrieval.

| Satellite/Sensor | Launch time | Spectral coverage (nm) | Spectral resolution (nm) | Swath (km) | Spatial resolution (km) | Overpass time |
|---|---|---|---|---|---|---|
| GOSAT | 2009.01 | 754-773 | 0.025 | 790 | 10.5 | 13:00 |
| OCO-2 | 2014 | 757-775 | 0.042 | 10.3 | 1.3×2.25 | 13:30 |
| OCO-3 | 2019.05 | 757-771 | 0.042 | ~10 | 1.6×2.2 | 13:30 |
| TanSat | 2016.12 | 758-778 | 0.044 | 20 | 2 | 13:30 |
| TECIS-1 | 2022.08 | 670-780 | 0.3 | 34 | 0.345 | 10:30 |
| SCIAMACHY | 2002.03 | 595-812 | 0.48 | 2800 | 30×240/60 | 10:30 |
| GOME-2 | 2007.01 | 590-790 | 0.5 | 1920 | 40×80/40 | 9:30 |
| TROPOMI | 2017.11 | 661-775 | 0.37 | 2600 | 3.5×5.5/7.5 | 13:30 |
| FLEX | >2025 | 500-780 | 0.3-2.0 | 300 | 0.3 | 10:00 |
| CO2M | 2026 | 747-773 | 0.12 | >250 | 2 | - |
| TanSat-2 | 2026 | 672-702 747-777 | 0.12 | 2900 | 2 | 13:30 |

**2 Materials**
**2.1 TanSat-2 satellite**
The second iteration of China's TanSat satellite, TanSat-2, operates in a critically inclined sun-synchronous orbit with an inclination
of approximately 116.6°. Positioned primarily above the Northern Hemisphere, the satellite is strategically placed to monitor
densely populated regions, such as Asia, North America, and Europe, where human activities are concentrated. By meticulously



adjusting various orbital elements, including inclination, semi-major axis, and eccentricity, TanSat-2 achieves a nodal precession
rate of roughly 0.98° per day eastward. This specific configuration preserves the sun-synchronous nature of the orbit, ensuring that
observations are made under consistent solar illumination at the same local time each day. The parameters of this orbit are detailed
in Table 2, which facilitates the simulation of the satellite's trajectory.
**Table 2.** TanSat-2 orbit parameters

| Orbit parameters | Value |
| --- | --- |
| Apogee altitude | 7443 km |
| Perigee altitude | 702 km |
| Inclination | 116.565 deg |
| Argument of perigee | 220 deg |
| Local time of ascending node | About 13:05 |

Equipped with four advanced payloads, TanSat-2 surpasses its predecessor in capability. These include the Ultra-Wide Swath
Hyperspectral imager for co-monitoring GHG and $NO_2$ (Uwhigo), which monitors greenhouse gases (GHGs) and nitrogen dioxide
($NO_2$) across a 1500 km swath with a 4 km resolution. The High-resolution Hyperspectral imager for monitor Hotspot emission
(H3imager) targets GHG emissions from localized sources with a 50 km swath and 500 m resolution. The Cloud and Aerosol
polarization hyperspectral imager (CAPHI) is designed to assess aerosol optical depth (AOD) and cloud coverage, while the dual-
band SIF imaging spectrometer (DuSIFIS) focuses on far-red and red SIF detection. The specifications for the DuSIFIS payload
are provided in Table 3.
**Table 3.** Preliminary specifications for the $O_2$ bands included in the TanSat-2 payloads.

| Bands | Spectral coverage (nm) | Spectral resolution (nm) | Spectral sampling interval (nm) | SNR | Spatial resolution | Swath |
| --- | --- | --- | --- | --- | --- | --- |
| $O_2$-A | 747-777 | | | 500 at $6.4*10^{19}$ photon/sec/m$^2$/sr/um | 2 km at 7000 km orbit altitude | 2900 km |
| | | 0.12 | 0.04 | | | |
| $O_2$-B | 672-702 | | | 780 at $1.6*10^{20}$ photon/sec/m$^2$/sr/um | | |

## 2.2 Simulation Experiments

### 2.2.1 Satellite radiative transfer simulation of TanSat-2

Assuming a Lambertian surface, the top-of-atmosphere (TOA) radiance observed by an instrument over vegetative targets can be
approximated as (Guanter et al., 2010; Liu et al., 2014;):
$$L_{TOA} \approx \frac{I_{sol}*\mu_0}{\pi} * \left[ \rho_0 + \frac{\rho_s*T_{\downarrow\uparrow}}{1-S*\rho_s} \right] + \frac{SIF*T_\uparrow}{1-S*\rho_s} \qquad (1)$$
where $I_{sol}$ is the extraterrestrial solar radiation, and $\mu_0$ denotes the cosine of the solar zenith angle (SZA). Reflectance from the
atmospheric path is indicated by $\rho_0$, and surface reflectance is denoted by $\rho_s$. The spherical albedo of the atmosphere is symbolized
by $S$. The total atmospheric transmittance, $T_{\downarrow\uparrow}$, encompasses both downward and upward transmission, with $T_\uparrow$ specifically
referring to the transmission from the surface to the sensor. The fluorescence signal ($SIF$), emitted at the top of the canopy (TOC),
should be excluded when modeling radiance for areas without vegetation.
The Soil Canopy Observation Photochemistry and Energy Flux (SCOPE) model developed by van der Tol et al. (2009) and is
capable of simulating vegetation canopy reflectance spectra and SIF signals under various conditions, including different canopy



structures and leaf biochemical properties. The model incorporates atmospheric radiative transfer functions derived from the
Moderate-resolution atmospheric TRANsmission (MODTRAN5) (Berk et al., 1998, 2000), to perform top-of-atmosphere radiance
forward simulations based on specific surface properties. Employing the most recent version of the MODTRAN interrogation
technique (MIT) (Verhoef and Bach, 2012; Verhoef et al., 2018), 18 spectral transfer functions were extracted and applied within
the SCOPE model's RTMo program to calculate interactions between the canopy surface and the atmosphere, as well as the solar
and sky irradiance spectra (Verhoef et al., 2018). These atmospheric parameters were then integrated with canopy reflectance and
SIF signals as outlined in Equation 1.
For the original MODTRAN 5 dataset, SR and SSI are approximately 0.005 nm. To simulate satellite observations, the original
spectral data underwent convolution and resampling techniques tailored to the specific SR and SSI of various bands, as detailed in
Table 3. Based on the configuration of the TanSat-2 satellite, we established two spectral channels with distinct spectral ranges.
The spectral response functions were modeled using Gaussian functions, which correlate with the SR (Zhao et al., 2022b; Zou et
al., 2022). This modeling process employs a filter kernel, K, defined by both the original and adjusted SR levels, to convolve the
initial 0.005 nm spectrum into the new spectral resolution. Consequently, the radiance for a given wavelength ($\lambda$) is calculated
using the following equations (Damm et al., 2011; Zou et al., 2022):
$$L_{TOA} \approx \frac{\int_{\lambda-\varepsilon}^{\lambda+\varepsilon} L_{TOA}(\hat{\lambda}) \cdot K(\lambda-\hat{\lambda}) d\hat{\lambda}}{\int_{\lambda-\varepsilon}^{\lambda+\varepsilon} K(\lambda-\hat{\lambda}) d\hat{\lambda}} \qquad (2)$$
$$K(\lambda-\hat{\lambda}) = \frac{2\sqrt{2ln2}}{\sqrt{2\pi}\sqrt{SR_d^2-SR_o^2}} \cdot \exp\left(-\frac{4(ln2)(\lambda-\hat{\lambda})^2}{SR_d^2-SR_o^2}\right) \qquad (3)$$
where $\lambda$ represents the center wavelength after undergoing spectral sampling, while $\hat{\lambda}$ corresponds to the wavelength in the original
dataset. $L_{TOA}(\hat{\lambda})$ denotes the TOA radiance output by MODTRAN and SCOPE simulations. The term $SR_o$ refers to the initial
spectral resolution (0.005 nm), and $SR_d$ corresponds to the spectral resolution relevant to the various channels. The parameter $\varepsilon$,
indicating the kernel's half-width in wavelength terms, was set to three times $SR_d$. For computational efficiency, a summation
approach was utilized in lieu of direct integration, with the differential wavelength interval, $d\hat{\lambda}$, fixed at 0.005 nm. This process
involved resampling the original spectra by selecting wavelength points ($\lambda$) at intervals defined by the SSI for each channel,
ensuring adequate coverage within the sensor's spectral coverage (SC).
Finally, random noise was incorporated into the data, quantified by the SNR, which varies with both radiance levels and
wavelength (Köhler et al., 2015; Zou et al., 2022).
$$SNR(Rad,\lambda) = SNR_{ref}\sqrt{\frac{Rad(\lambda)}{Rad_{ref}}} \qquad (4)$$
where $SNR_{ref}$ represents the reference SNR at the reference radiance level $Rad_{ref}$.
**2.2.2 Spectral simulation dataset**
In the context of vegetative scenes, a total of 1008 cases were examined, incorporating combinations of diverse leaf biochemical
characteristics, canopy structures, and geometric conditions, as illustrated in Table 4. For non- vegetation targets, ten distinct
reflectance spectra representing snow and bare soil surfaces were sourced from the Image Visualization Environment (ENVI)
spectral library (Clark and Swayze, 1995). A total of 1280 atmospheric and observational conditions were simulated (Table 4),
with the observational conditions aligning with those of the SCOPE model. Ultimately, 12,800 non-vegetated spectra and 161,280
vegetative spectra were generated as training and test datasets.



**Table 4.** Input parameters used in the MODTRAN and SCOPE.

| Parameters of MODTRAN5 | Value |
| --- | --- |
| Atmospheric temperature profile | Middle latitude summer/winter |
| Aerosol optical thickness at 550 nm | 0.05, 0.12, 0.2, 0.3, 0.4 |
| Vertical water vapor column (g cm$^{-2}$) | 0.5, 1.5, 2.5, 4 |
| Surface altitude (km) | 0.01, 0.05, 1, 2 |
| Solar zenith angle (degree) | 15, 30, 45, 70 |
| View zenith angle (degree) | 0, 16 |
| Parameters of SCOPE | Value |
| Leaf area index (LAI) | 0.5, 1, 2, 3, 4, 5, 7 |
| Fluorescence quantum efficiency (fqe) | 0.01, 0.02, 0.04 |
| Chlorophyll content (Cab) (ug cm$^{-2}$) | 20, 30, 40, 50, 60, 80 |
| Solar zenith angle (degree) | 15, 30, 45,70 |
| View zenith angle (degree) | 0, 16 |

Figure 1 presents the simulated spectrum with an SR of 0.12 nm and an SSI of 0.04 nm. The displayed solar-induced fluorescence (SIF) spectra exhibit characteristic double peaks in the red and far-red bands, around 685 nm and 740 nm, respectively. Additionally, the spectra distinctly reveal the absorption features of the Earth's atmosphere, as well as the Fraunhofer lines.

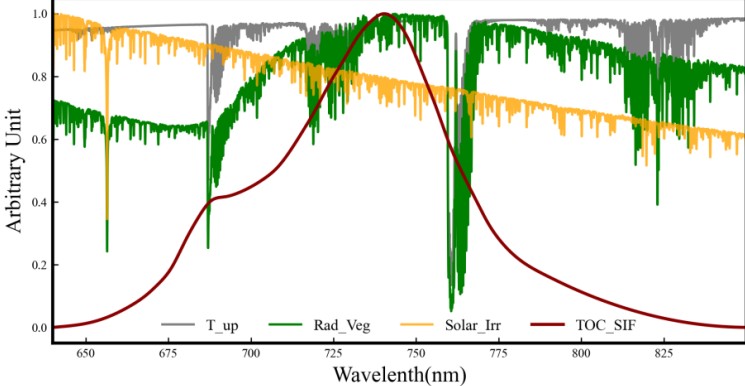

**Figure 1.** A collective set of normalized, simulated spectra at the TOA over vegetated surfaces (Rad_Veg), together with the incident solar radiance that arrives at the TOC (Solar_Irr), upward transmittance of the atmosphere (T_up), and SIF at the TOC (TOC_SIF). These spectra are presented with an SR of 0.12 nm and an SSI of 0.04 nm derived from SCOPE and MODTRAN 5 models.

**2.2.3 End-to-end orbit simulation dataset**

TanSat-2 boasts a spatial resolution of 2 km paired with a swath width of 2,900 km, enabling it to offer high spatio-temporal resolution and comprehensive observations. Its expansive swath ensures nearly global coverage each day. Utilizing the J2 orbit perturbation model and TanSat-2's orbital elements, we simulated its orbit—a critically inclined, sun-synchronous trajectory. This simulation facilitated the assessment of global surface observations to model the worldwide spatiotemporal distribution of data collected by TanSat-2. Under typical acquisition scenarios, global simulations were conducted, harnessing the satellite's spatiotemporal resolution to simulate the TOA radiance spectra received by TanSat-2. The aim is to explore the spatiotemporal capabilities of TanSat-2 in retrieving SIF from its global observations.




For our study, mid-June was selected due to the peak greenness of vegetation in the Northern Hemisphere, which corresponds
with the greatest global variability in vegetation cover. The end-to-end simulation of TOA radiance, in line with Equation 1 and
following the spectral simulation process described in Section 2.2.2, combined the SCOPE and MODTRAN models to generate
the reflectance and SIF spectra, as well as atmospheric functions. Limitations in globally representative data for vegetation and
atmospheric parameters introduce some discrepancies in the end-to-end simulation processes compared to the spectral simulation,
as depicted in Fig. 2.

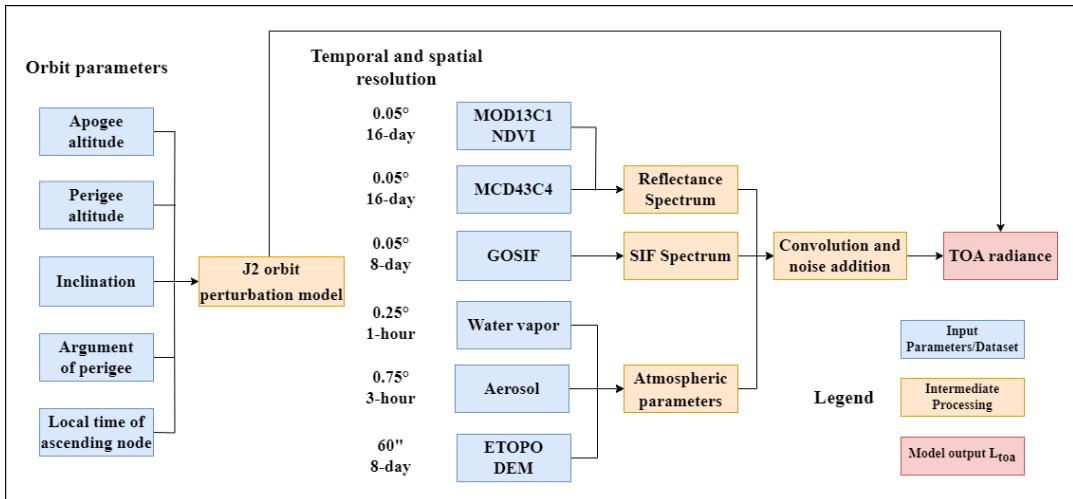

**Figure 2.** Flowchart of the end-to-end orbit simulations.
Reflectance spectra calculations utilized the green, red, and near-infrared bands from the Moderate Resolution Imaging
Spectroradiometer (MODIS) MCD43C4 product (Huete et al., 2002). To accurately reflect real-world conditions, different
simulation strategies for vegetation and non-vegetation surfaces were implemented. Global screening employed NDVI data from
the MODIS MOD13C1 product (Huete et al., 2002), using a threshold of 0.2 to differentiate between non-green and green
vegetation. Non-vegetation reflectance spectra were fitted using quadratic polynomials. Meanwhile, vegetation reflectance spectra,
due to their complexity in these bands, were simulated using the data from Section 2.1. A singular value decomposition (SVD) was
performed, and the first two basis vectors were extracted to fit the spectral data, as illustrated in Fig. 3, with interpolation across
the 640-800 nm range. For SIF spectra, the global OCO-2 SIF dataset (GOSIF) (Li et al., 2018) was used to map the global SIF
distribution, filling data gaps through interpolation and depicting SIF spectral shape with a typical profile shown in Fig. 1.
Datasets representing the global distribution of atmospheric conditions were assembled, incorporating water vapor data sourced
from the ERA5, the fifth generation reanalysis of global climate and weather by the European Centre for Medium-Range Weather
Forecasts (ECMWF) (Hersbach et al., 2020), and aerosol information from ECMWF's Atmospheric Composition Reanalysis 4
(EAC4), provided by the Copernicus Atmosphere Monitoring Service (CAMS) (Inness et al., 2019). Additionally, the 60-arcsecond
resolution DEM data from Earth TOPOgraphy (ETOPO), furnished by the National Oceanic and Atmospheric Administration
(NOAA) (Amante and Eakins, 2009), was resampled to a spatial resolution of 0.02 degrees to align with the satellite payload
specifications. To accommodate the atmospheric conditions data, we expanded the range of atmospheric parameters outlined in
Table 4, with further details provided in Table 5. These data are essential for calculating parameters such as $\rho_0$, $S$, $T_{\downarrow\uparrow}$, and $T_\uparrow$ in
Equation 1, which are crucial for simulating TOA radiance. To optimize simulation times, we eschewed the use of MODTRAN 5
for per-pixel atmospheric parameter simulation, opting instead for a random forest model. We sampled 10% of the atmospheric
conditions from Table 5—a total of 6,912 data points—and input them into MODTRAN 5 to simulate key atmospheric parameters.



These parameters were then used to train the random forest model, which was subsequently employed to simulate the atmospheric
parameters for each pixel. These simulations were incorporated into Equation 1 to compute TOA radiance. Additional processes,
including convolution and noise addition, were applied. Figure 4 illustrates a pseudo-color image synthesized using the near-
infrared, red, and green bands of MCD43C4, displaying the fitted reflectance and TOA radiances across seven representative
geomorphic areas.

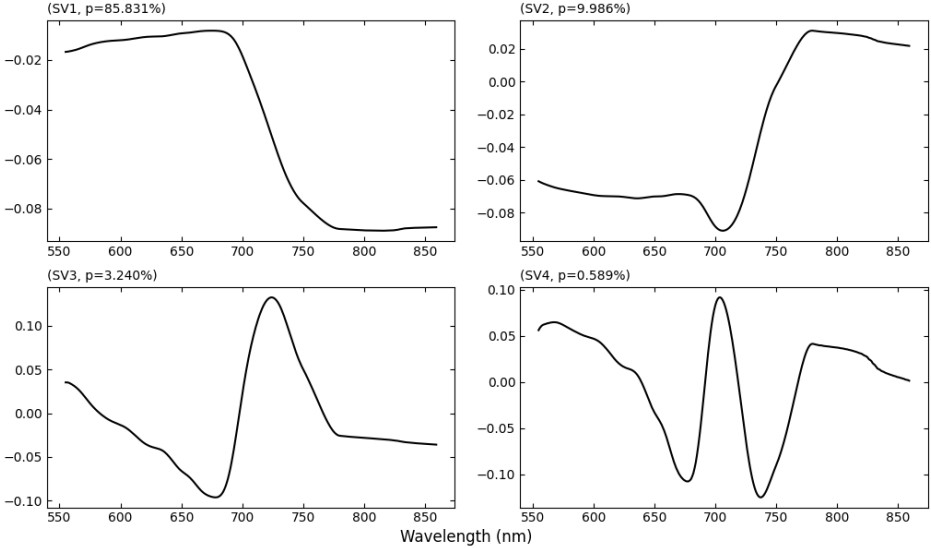

**Figure 3**. The first four singular vectors from the singular value decomposition of the reflectance training dataset.
**Table 5.** Look-up table (LUT) of MODTRAN 5 used to train the atmospheric parameter random forest model

| Parameters of MODTRAN5 | Value |
|---|---|
| Atmospheric temperature profile | Tropical, middle latitude summer/winter, subarctic summer/winter |
| Aerosol optical thickness at 550 nm | 0.05, 0.12, 0.2, 0.3, 0.4, 0.6, 1.0, 2.0, 3.0, 4.0 |
| Vertical water vapor column (g cm$^{-2}$) | 0.5, 1.5, 2.5, 4.0, 6.0, 8.0 |
| Surface altitude (km) | 0.01, 0.05, 1, 2, 3, 4, 5, 6 |
| Solar zenith angle (degree) | 15, 22, 30, 37, 45, 52, 60, 70 |
| View zenith angle (degree) | 0, 8, 16, 25 |





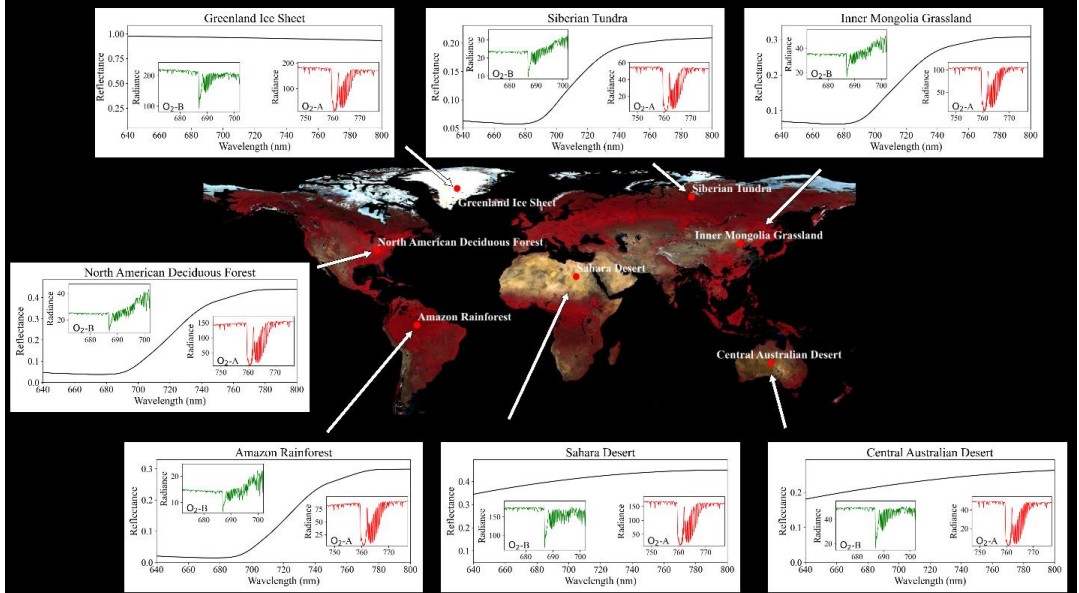

**Figure 4**. Global reflectance pseudo-color composite and the BOA reflectance along with simulated TOA radiance for two channels in typical regions.

Moreover, we adopted strategies similar to those utilized by Du et al. (2018) to create datasets as training samples for the data-driven method. Our selection criteria targeted non-vegetation surfaces, such as bare soil and snow, to minimize biases and uncertainties. The SZA and view zenith angle (VZA) were restricted to less than 75° and 60°, respectively, ensuring that the SZA within the training dataset were representative. We used the MODIS MCD43C4 BRDF-adjusted reflectance product to identify non-vegetation surfaces. As demonstrated by Guanter et al. (2012), the criterion RNIR < RSWIR effectively differentiates bare soil from other surface types. Bright soil pixels, characterized by high reflectivity, were identified using the MODIS data, specifically when the reflectance in band 6 surpassed that in band 2 and exceeded a threshold value of 0.2. Additionally, snow-covered areas were delineated employing the Normalized Difference Snow Index (NDSI) as defined by Salomonson and Appel (2004). A pixel was identified as snow-covered when its NDSI value exceeded 0.4, with further constraints on reflectance values in the near-infrared (band 2) above 0.11 and in band 4 above 0.10, following the criteria established by Riggs et al. (2006).

To ensure the training dataset's representativeness and variability, spectra were selected to be uniformly distributed across the globe. Consequently, 400,000 spectra were randomly sampled to achieve this diversity.

**3 Retrieval method for TanSat-2 simulation dataset**

**3.1 Data-Driven SIF Retrieval Algorithm**

Singular value decomposition (SVD) technology, akin to Principal Component Analysis, is extensively utilized for solving linear equations and is pivotal in statistical analysis and reducing dimensionality in large datasets (Du et al., 2018). In models driven by data, the initial segment of Equation (1), which excludes fluorescence effects, integrates high-frequency and low-frequency components. As delineated by Guanter et al. (2012, 2013), high-frequency components, stemming from atmospheric absorption influenced by solar and terrestrial sources, are extracted from non-vegetation datasets through SVD and reconstructed using a select set of characteristic spectra. Conversely, the low-frequency components, representing atmospheric scattering (denoted as $\rho_0$





and S) and surface reflectance ($\rho_s$), are approximated using polynomial wavelength functions ($\lambda$). Moreover, the shape of SIF
spectra is typically modeled through a normalized Gaussian function ($h_f$).
$$h_f = \exp\left(-\left[\frac{-(\lambda-\lambda_0)^2}{2\sigma_h^2}\right]\right) \tag{5}$$
where $\lambda_0$ is the peak emission wavelength of SIF spectral in the far-red and red bands, and the value of $\sigma_h$ determines the shape of
SIF spectral. Therefore, equation (1) can be rewritten as:
$$L_{TOA} = v_1 \cdot \sum_{j=0}^{n_p}(\beta_j\lambda^j) + \sum_{k=2}^{n_v}(\gamma_k v_k) + F_s \cdot h_f \cdot T_\uparrow^e \tag{6}$$
where $\beta$ and $\gamma$ represent the coefficients vectors to be determined, $n_p$ is the polynomial order, v refers to the singular vectors of the
high-frequency components, with their number given by $n_v$. $T_\uparrow^e$ is the effective upward transmittance to represent the atmospheric
transmittance from the surface to the sensor. $T_\uparrow^e$ can be estimated before retrieval (Köhler et al., 2015):
$$T_\uparrow^e = exp\left[lnT_{\downarrow\uparrow}^e \cdot \frac{sec(\theta_v)}{sec(\theta_0)+sec(\theta_v)}\right] \tag{7}$$
where $T_{\downarrow\uparrow}^e$ represents the effective two-way atmospheric transmittance, obtained by normalizing the TOA reflectance using low-
order polynomials. $\theta_0$ and $\theta_v$ is the SZA and VZA, respectively.
For spectral characterization of SIF, the parameters $\lambda_0$ and $\sigma_h$ for the far-red band are set at 740 nm and 21 nm, respectively.
The red band utilizes a combination of two Gaussian functions to depict a more intricate spectral shape, with $\lambda_0$ being 740 nm and
685 nm, and $\sigma_h$ being 21 nm and 10 nm, respectively (Joiner et al., 2016; Zou et al., 2022). This spectral region encapsulates
numerous solar Fraunhofer lines and atmospheric absorption features, enhancing the retrieval capabilities of SIF. Specifically, the
spectral domain of TanSat-2 encompasses absorption lines such as the $O_2$-A at 758-772 nm and the $O_2$-B at 682-692 nm, which are
integral for retrieving far-red and red SIF, respectively (Joiner et al., 2013; Guanter et al., 2015). Surrounding solar Fraunhofer and
atmospheric absorption lines also play a crucial role in the SIF retrieval process in satellites with refined spectral resolution
(Frankenberg et al., 2011; Joiner et al., 2011). As part of a semi-empirical approach, the performance of data-driven algorithms
heavily relies on the empirical parameters used in the model. To optimize these parameters, different window settings were applied
for each channel. The window settings for the far-red band were 747-758 nm, 759-772 nm, and 747-777 nm; for the red band, they
were 672-686 nm, 682-697 nm, and 672-702 nm. Moreover, the permissible ranges for the parameters $n_p$ (0-7) and $n_v$ (1-50) were
established for SIF retrieval (Köhler et al., 2015; Zou et al., 2022). Ultimately, only the parameters $a_i$, $\beta_j$, $\gamma_k$, and $F_S$ remained as
variables, with $F_S$ determined through resolving the linear least squares problem.
SVD efficiently transforms a large set of correlated variables into a streamlined set of uncorrelated components, known as
singular vectors. These vectors are strategically arranged such that each successive vector accounts for progressively less signal
variability, enabling a hierarchical representation of data. By leveraging the principal singular vectors, we can reconstruct similar
signals and effectively filter out noise. The implementation of SVD was carried out on the training dataset from spectral simulations.
Figure 5 illustrates the first six basis vectors for two distinct channels. Each subplot also quantifies the explained variance
associated with each basis vector in the simulations. Predominantly, the spectral variations within the fitting window arise from
Fraunhofer lines and atmospheric absorption features. It is evident that the initial set of singular vectors encapsulates the majority
of the spectral variance across all simulations, while none of the vectors correspond to the SIF spectral shape.





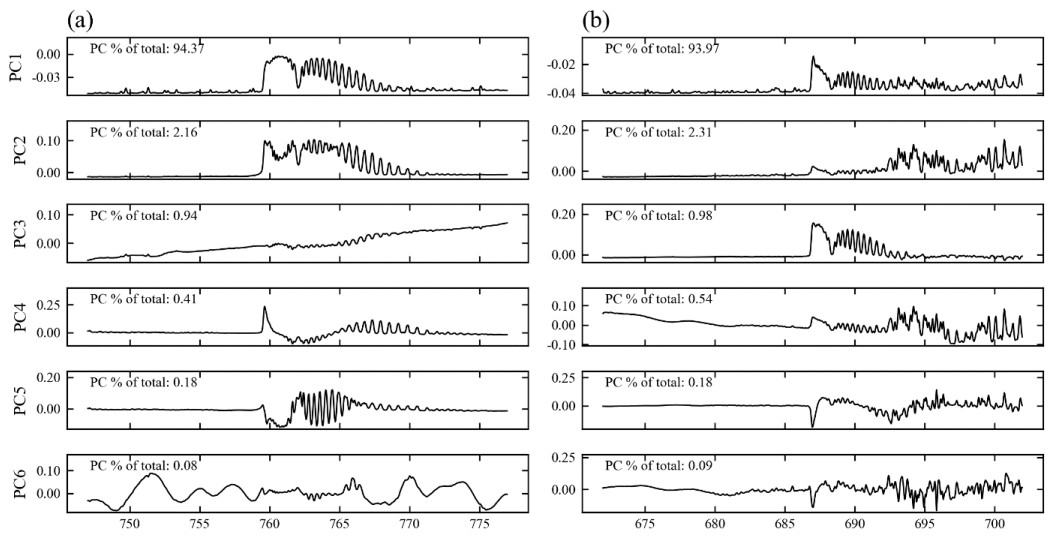

**Figure 5**. The first 6 bias vectors from the SVD of the spectral simulation training dataset in two spectral ranges: (a) 747–777 nm and (b) 672–
702 nm.
**3.2 Metrics for Evaluating Accuracy**
For the simulated SIF intensities at 740 nm and 685 nm, we designated these values as the true SIF signals ($SIF_{true}$) for each channel,
respectively. To evaluate the accuracy of the spectral retrieval, we compared the retrieved SIF values ($SIF_{retrieved}$) at these
wavelengths with $SIF_{true}$. The analysis included calculating the root-mean-squared error (RMSE) to quantify the retrieval precision.
Additionally, several statistical metrics were employed to further assess the retrieval performance, such as the slope and intercept
from the linear regression between $SIF_{retrieved}$ and $SIF_{true}$, alongside the coefficient of determination ($R^2$). The adjusted SIF values
($SIF_{corr}$), intended to rectify systematic biases inherent in data-driven retrieval methods, were computed as per Equation (8) (Du et
al., 2020; Zou et al., 2022).
$$SIF_{corr} = \frac{SIF_{retrieved} - intercept}{slope} \qquad\qquad (8)$$
where the intercept and slope represent the parameters of the linear relationship between $SIF_{retrieved}$ and $SIF_{true}$, expressed as
$SIF_{retrieved} = SIF_{true} \times slope + intercept$. The corrected RMSE (RMSE*) was calculated by comparing $SIF_{true}$ with $SIF_{corr}$.
**3.3 Quality filtering rules for global SIF composites**
Global composites of SIF were generated by averaging retrieval data across a grid with a resolution of 0.05° by 0.05° over a 1-day
period. Before averaging, the retrieval data underwent a filtering process based on established quality criteria (Du et al., 2018;
Guanter et al., 2021). The criteria include:
(1)  Land surface.
(2)  SZA of <75° and the VZA of <60°.
(3)  $\chi^2_{red}$ estimates inside the 95% range of expected values.
where Chi-square ($\chi^2$) test is a statistical method used to evaluate how well a model fits observed data by measuring the difference
between observed and fitted data. If the fitted radiances deviate significantly from the observed radiances, it indicates a poor fit.
The reduced Chi-square ($\chi^2_{red}$) is calculated by dividing the Chi-square value by the degrees of freedom.





**4 Results**
**4.1 Optimization of empirical parameters in the data-driven algorithm using spectral simulations**
**4.1.1 Impact of empirical parameters on SIF retrievals**
To refine the parameterization of the algorithm that guides the TanSat-2 SIF retrieval, we have conducted a quantitative analysis
based on spectral simulations. This study evaluates the impact of various empirical parameters utilized in data-driven algorithms
on the retrieval outcomes. Parameters investigated include the configuration of the fitting windows, the number of eigenvectors
($n_v$), and the polynomial order ($n_p$).
In our analysis, we implemented differing window settings across two spectral channels, as depicted in Fig. 6. A comparative
analysis reveals substantial variations in retrieval performance among the window settings. Notably, the optimal retrieval windows
for both channels exclusively utilized atmospheric windows for retrieval, achieving root mean square errors (RMSEs) of 0.24 and
0.19 mW m$^{-2}$ sr$^{-1}$ nm$^{-1}$, respectively. Wider fitting windows tend to encompass atmospheric absorption bands, offering a plethora
of usable spectral data. However, this approach necessitates precise simulation of atmospheric upwelling transmittance and
complicates the modeling of vegetation surface reflectance and SIF spectral shape, particularly in the red band. Consequently, a
narrower atmospheric window band frequently serves as a more advantageous choice for the spectral parameters of TanSat-2.

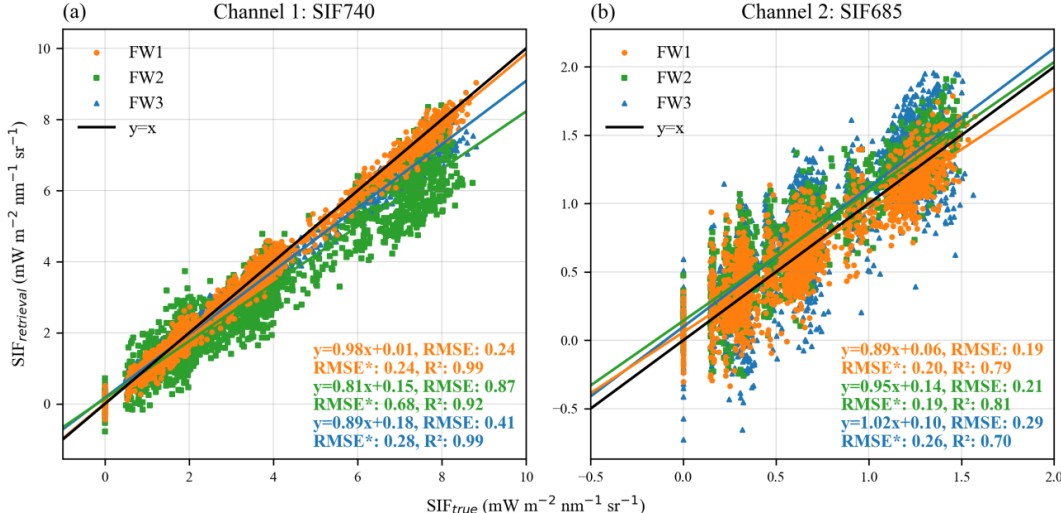

**Figure 6**. Retrieval using simulated datasets of two channels with different fitting windows: the windows for channel 1 are 747-758 nm, 759-772
nm, and 747-777 nm; the windows for channel 2 are 672-686 nm, 682-697 nm, and 672-702 nm. For demonstration purposes, only a randomly
selected portion of the data is used to draw the scatter plot.
The relationship between the number of eigenvectors and RMSEs was examined within the context of the optimal fitting window,
as illustrated in Fig. 7. The analysis indicates that RMSEs initially decrease, subsequently increase, and ultimately stabilize as the
number of eigenvectors grows. Optimal $n_v$ values were determined to be 6 for channel 1 and 4 for channel 2.



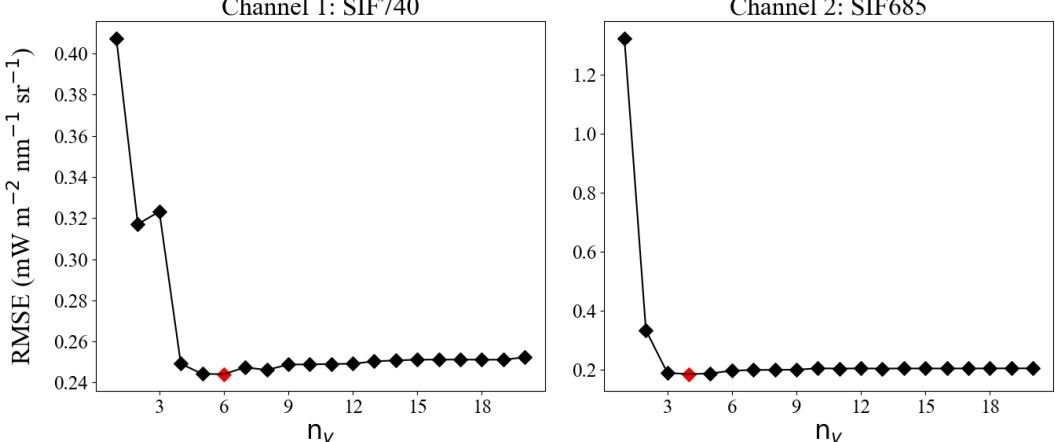

**Figure 7.** RMSE of SIF retrieval in the optimal fitting windows using varying numbers of eigenvectors. The red point refers to the optimal $n_v$.
Data-driven models commonly employ polynomials to approximate the shape of reflectance and atmospheric scattering within
the fitting window. Typically, a broader fitting window necessitates the use of higher-order polynomials. The red band, positioned
near the vegetation's red edge and heavily influenced by chlorophyll absorption, exhibits notably complex reflectance patterns.
Thus, higher polynomial orders are generally required for the red band compared to the far-red band. An analysis utilizing the
optimal window revealed that RMSE values initially decrease and then increase with the escalation of polynomial order, as detailed
in Fig. 8. This pattern arises because low polynomial orders fail to accurately model the reflectance spectral shape, whereas
excessively high orders induce overfitting, obstructing the detection of faint SIF signals within the upstream radiance. After
surpassing a certain threshold, RMSE stabilizes at a high value. For channels 1 and 2, the optimal polynomial orders were found
to be 2 and 4, respectively.

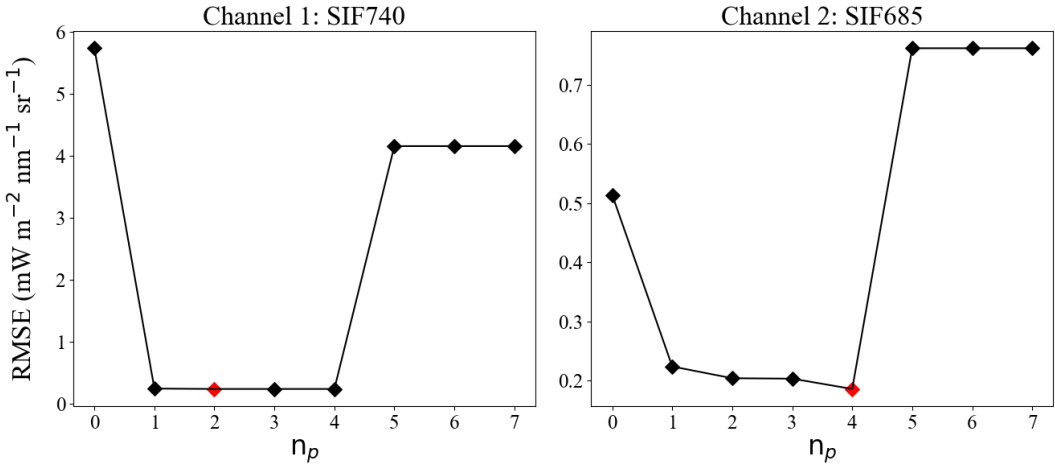

**Figure 8.** RMSE of SIF retrieval in the optimal fitting windows using a varying polynomial order. The red point refers to the optimal $n_p$.
**4.1.2 SIF retrievals using optimized empirical parameters**
SIF retrieval was performed using the spectral simulation of the TanSat-2 payload configuration to evaluate its retrieval potential.
We assessed the retrieved SIF values ($SIF_{retrieval}$) against the actual values ($SIF_{true}$) under varied atmospheric conditions, leaf





biochemical characteristics, canopy structure, and geometrical arrangements. The results presented in Fig. 9 illustrate the SIF
retrievals for TanSat-2, utilizing the optimally derived empirical parameters.

3       For the two spectral channels examined, the scatter plot distributions align closely with the 1:1 line, indicating high retrieval

accuracy. In channel 1, the data-driven algorithm was optimized with settings that included a spectral fitting window from 747–
758 nm, a second-order polynomial fitting, and six eigenvectors, yielding an RMSE of 0.24 mW m$^{-2}$ sr$^{-1}$ nm$^{-1}$. Channel 2 employed
a fourth-order polynomial and four eigenvectors within a 672–686 nm window, achieving an RMSE of 0.19 mW m$^{-2}$ sr$^{-1}$ nm$^{-1}$.
Furthermore, the correlation between the RMSE and its normalized counterpart (RMSE*) suggests minimal systematic errors in
the retrieval methodology.

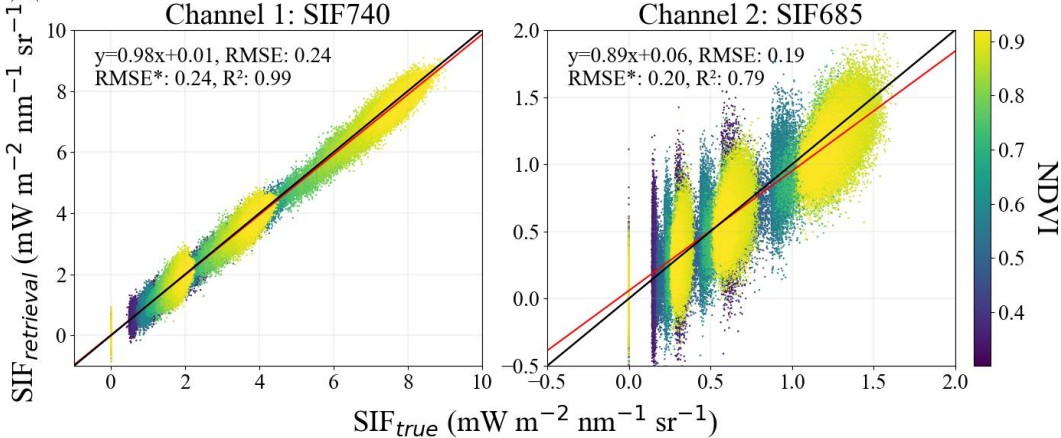

**Figure 9.** SIF retrievals of spectral simulation using two channels with optimal parameter settings. The black line denotes 'y = x', while the red
one denotes the linear fitting line.
To illustrate the forward model fit, TOA radiance spectra for the ranges 747–758 nm and 672–686 nm were simulated using both
the standard forward model and a variant excluding SIF. Comparison of these spectra against actual TOA radiance measurements
yielded two spectral residuals, depicted in Fig. 10. Models incorporating SIF showed significantly lower spectral residuals,
particularly around the Fraunhofer lines. Additionally, in the forward model that includes SIF, the spectral residuals across both
channels were generally minimal, underlining the efficacy of the SVD method in fitting data and accurately reconstructing surface
spectral shapes absent of SIF signals.





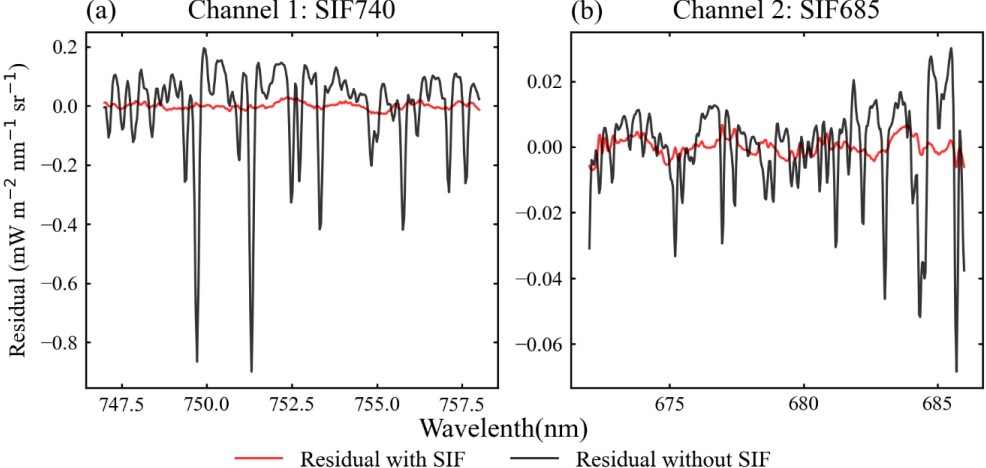

**Figure 10**. Spectral residuals between the fitted and observed TOA radiances, shown for cases where the model includes SIF (red) and excludes
SIF (black) in two spectral ranges: (a) 747–758 nm and (b) 672–686 nm. (a) and (b) correspond to channels 1 and 2.
**4.2 Evaluation of SIF retrievals based on end-to-end orbit simulations**
In Section 4.1, we evaluated the impact of empirical parameters on SIF retrieval accuracy using a data-driven model and established
an optimal algorithm for the two channels on TanSat-2, based on spectral simulations detailed in Section 2.2.2. This section also
discusses how retrieval errors from a single pass of TanSat-2 propagate into the spatiotemporal composite. Fig. 11 illustrates the
global distribution of observations measured by TanSat-2 within a 0.05° global grid over a single day. Despite the fine resolution,
the satellite's high spatial resolution and wide swath enabled a substantial volume of observations, approximately 5-7 times greater
than that achieved by TROPOMI. These payload characteristics of TanSat-2 significantly enhance the reliability of SIF retrievals.

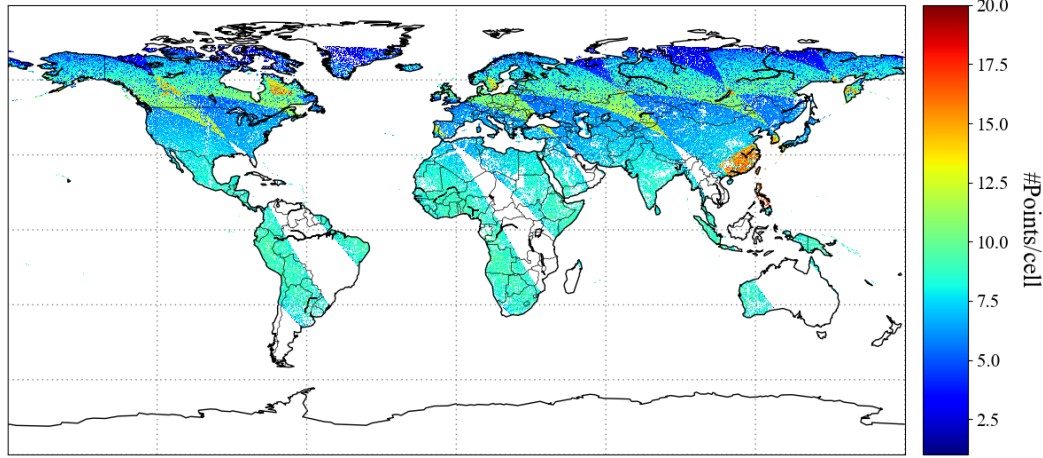

**Figure 11.** Global map showing the total number of TanSat-2 observations. The simulations represent daily averages for June, with a grid
resolution of 0.05°.
The aggregated SIF retrieval results for two bands and the global observations of SIF (GOSIF) over a 1-day period, displayed
in Fig. 12 with a grid size of 0.05°, demonstrate a strong correlation. High SIF values are noted in densely vegetated regions, with
the highest global SIF signals occurring in the eastern United States. Conversely, SIF values near zero are observed in regions such



as Greenland, the Sahara Desert, and much of Australia. For the analysis, GOSIF values at 757 nm were converted to 740 nm and
685 nm to represent the actual SIF of the two channels. This conversion involved multiplying the 757 nm SIF values by factors of
1.48 and 0.54, respectively, which align with the SIF spectral shape used in the end-to-end orbit simulated dataset (refer to Fig. 1).
A comparison of the retrieved SIF with the true SIF shows discrepancies generally less than 0.15 mW m$^{-2}$ sr$^{-1}$ nm$^{-1}$, as illustrated
in Fig. 13. The R$^2$ values for the two channels are 0.88 and 0.61, respectively, with corresponding RMSE values of 0.082 and 0.061.
These RMSE values represent approximately 5.4% and 11.1% of the globally observed SIF peaks, which are considered acceptable
error thresholds for SIF synthesis (Fig. 14). Moreover, we evaluated the results for 4-day and 8-day composites, shown in Figs. A2
and A4. As the number of observations increases, R² improves significantly, while RMSE decreases, further demonstrating the
instrument's performance onboard the TanSat-2 satellite. It should be noted that although the AOD product used partially accounts
for cloud impacts, it does not explicitly model cloud contamination or apply cloud fraction products for screening. Consequently,
the results in Figs. 12–14 assume clear-sky conditions. Additionally, these simulations do not incorporate rotational Raman
scattering (RRS), as the filter only considers data with a solar zenith angle (SZA) less than 75°, where the RRS effect is expected
to be minimal.





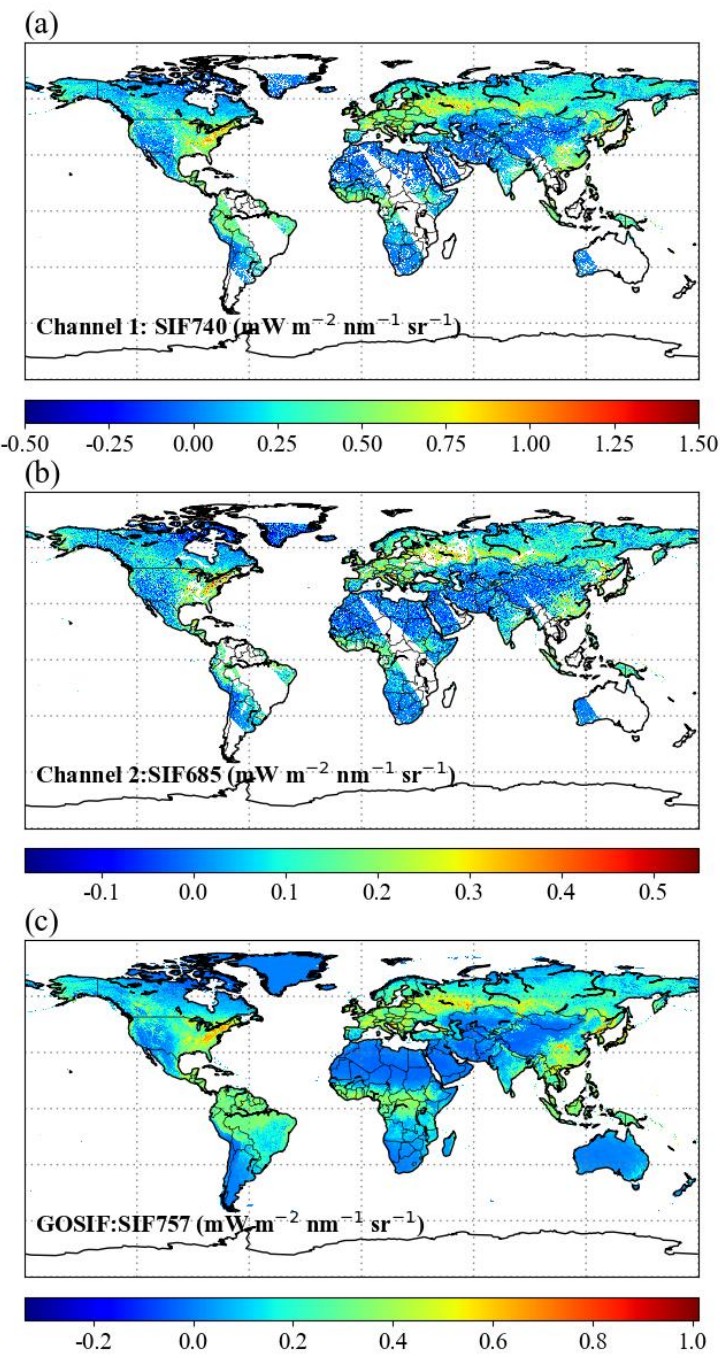

2    **Figure 12.** Global maps of mean SIF for each of the two channels, with GOSIF serving as the reference for true SIF, under the same conditions

3    as Fig. 11. The SIF range is scaled according to the proportion of the SIF spectrum used.



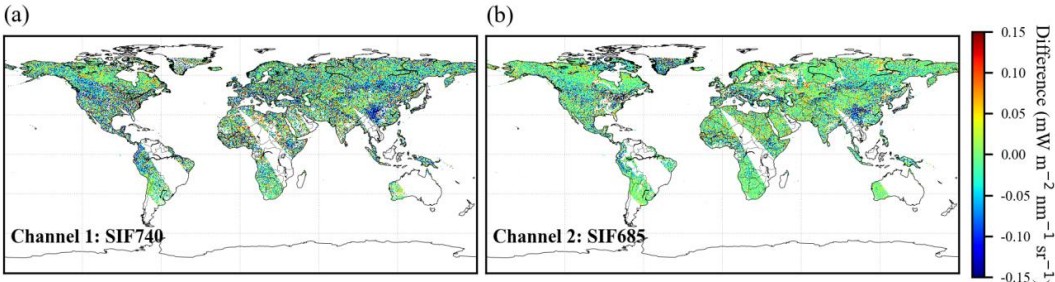

**Figure 13.** Differences between the retrieved SIF and the true SIF for two channels, under the same conditions as Fig. 11.

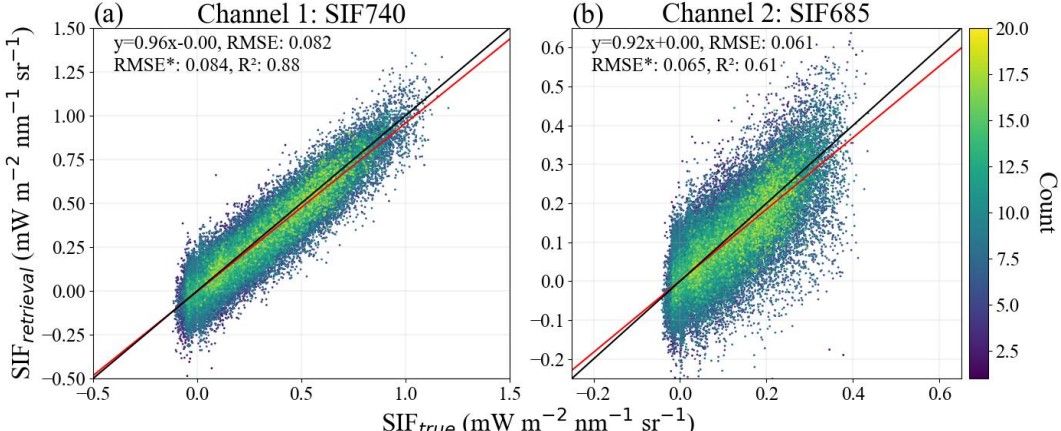

**Figure 14.** Global simulations of SIF retrieval from two channels, under the same conditions as Fig. 11. The black line denotes 'y = x', while the
red one denotes the linear fitting line.
**5 Discussion**
**5.1 Representativeness of simulations to satellite observations**
For the end-to-end orbit simulation, we modeled TanSat-2's Earth observations based on its orbital parameters, using data from
various sources to represent the global distribution of surface reflectance, SIF, and atmospheric conditions. Although the simulation
is driven by satellite datasets, it cannot fully capture the complexities present in actual satellite measurements, such as instrument
artifacts or RRS effects. In the simulation, radiances are initially generated monochromatically through scattering, and then
convolved with the instrument response function. The instrument response function is modeled using a Gaussian profile. For
simulating random noise, a simplified signal-to-noise ratio (SNR) model (Equation 4) was applied. Zou et al. (2022) compared the
simulated spectra and SNR with actual satellite measurements, and the results demonstrated high consistency between the two,
validating the accuracy of the SNR model and the spectral filtering process. Thus, it is feasible to assess the SIF retrieval capability
of TanSat-2 based on the simulated dataset.
Additionally, while the AOD dataset partially incorporates the influence of clouds—particularly in areas of high value—we did
not explicitly simulate cloud contamination or utilize cloud cover products for screening purposes. The satellite's observations
inherently reflect the impact of clouds on the radiance signal. The establishment of various cloud screening criteria involves a
trade-off between mitigating the clear-sky bias inherent in satellite observations and optimizing the accuracy of SIF retrieval



(Frankenberg et al., 2014b). Stringent cloud screening measures can lead to a reduction in the overall number of usable
measurements, consequently producing a noisier output map. This paper aims to investigate the benefits derived from the TanSat-
2 sensor's superior spectral observation capabilities and its high temporal and spatial resolution in SIF retrieval. As such, our end-
to-end orbital simulation operates under the assumption of clear sky conditions, deliberately excluding the potential impacts of
cloud cover. Furthermore, the CAPHI instrument aboard the satellite will supply data on AOD and cloud coverage, enhancing our
understanding of atmospheric conditions and cloud dynamics, while also providing supplementary data for future SIF retrieval
efforts with TanSat-2.
**5.2 Advancing global vegetation dynamics insights through TanSat-2 observations**
The TanSat-2 mission is equipped with a SIF imaging spectrometer that continuously gathers high spectral resolution data across
the $O_2$-A and $O_2$-B bands, boasting excellent SNR. The precise retrieval of red and far-red SIF is vital for advancing our
comprehension of plant photosynthesis and global carbon cycles. Moreover, the satellite's wide-swath and high spatial resolution
capabilities promise significant improvements in the informational content of SIF data, especially in regions characterized by
fragmented vegetation, thereby facilitating more frequent and valid observations.
The integration of SIF signals from distinct spectral bands offers unique advantages for monitoring vegetation health and
photosynthesis (Liu et al., 2017). Notably, the majority of SIF emissions stem from photosystem II (PSII) protein complexes
involved in photosynthesis, whereas photosystem I (PSI) predominantly contributes to far-red fluorescence, with its influence
increasing at longer wavelengths and typically remaining unaffected by biochemical factors. In contrast, the contributions from
PSII are modulated by physiological regulation, leaf structure, and chemical composition (Verrelst et al., 2015). Ač et al. (2015)
conducted a meta-analysis that concluded canopy-level red and far-red SIF diminishes under water stress, while the ratio of red to
far-red fluorescence increases in response to nitrogen deficiency. Red and far-red SIF can serve as early indicators of both water
stress and recovery (Daumard et al., 2010; Xu et al., 2018). By utilizing high-precision spectral data from TanSat-2 for dual-band
SIF retrieval, the capacity to detect vegetation physiological parameters is significantly enhanced, offering substantial advantages
for monitoring vegetation health. Furthermore, Verrelst et al. (2015) demonstrated that peaks in red SIF emission are strongly
influenced by carboxylation capacity (Vcmo), which correlates with photosynthetic capacity. Consequently, employing dual-band
emission fluxes, rather than relying solely on a single far-red band, proves more effective for correlating SIF with photosynthetic
metrics such as gross primary productivity (GPP).



**6 Conclusion**
The TanSat-2 satellite is specifically designed for global carbon inventory verification. It features a wide swath of 2900 km and
high spatial resolution of 2 km at an orbit altitude of 7000 km, providing near-daily global baseline coverage. Additionally, it is
equipped with a wide spectral range and high spectral specifications, enabling dual-band SIF retrievals with unprecedented
accuracy. This paper employs spectral simulations to optimize empirical parameters for data-driven SIF retrieval in two channels
of the TanSat-2 satellite. Using the optimized data-driven algorithm, the RMSEs of SIF retrievals for the two channels are 0.24 m
and 0.19 mW m$^{-2}$ sr$^{-1}$ nm$^{-1}$, respectively. Furthermore, the potential of the forthcoming TanSat-2 satellite for global SIF monitoring
is evaluated through end-to-end simulations at a global scale. Comparisons between daily and 0.05° global composites of dual-
band SIF retrievals and GOSIF indicate strong agreement, with R$^2$ values of 0.88 and 0.61, and RMSE values of 0.082 and 0.061
mW m$^{-2}$ sr$^{-1}$ nm$^{-1}$ for the SIF retrievals at the two bands. Therefore, TanSat-2 presents significant opportunities for SIF retrieval at
both red and far-red bands, offering high resolution, precision, and frequency of coverage, which is crucial for a comprehensive
understanding of global vegetation photosynthetic activities and the terrestrial carbon cycle.





1    **Appendix A**

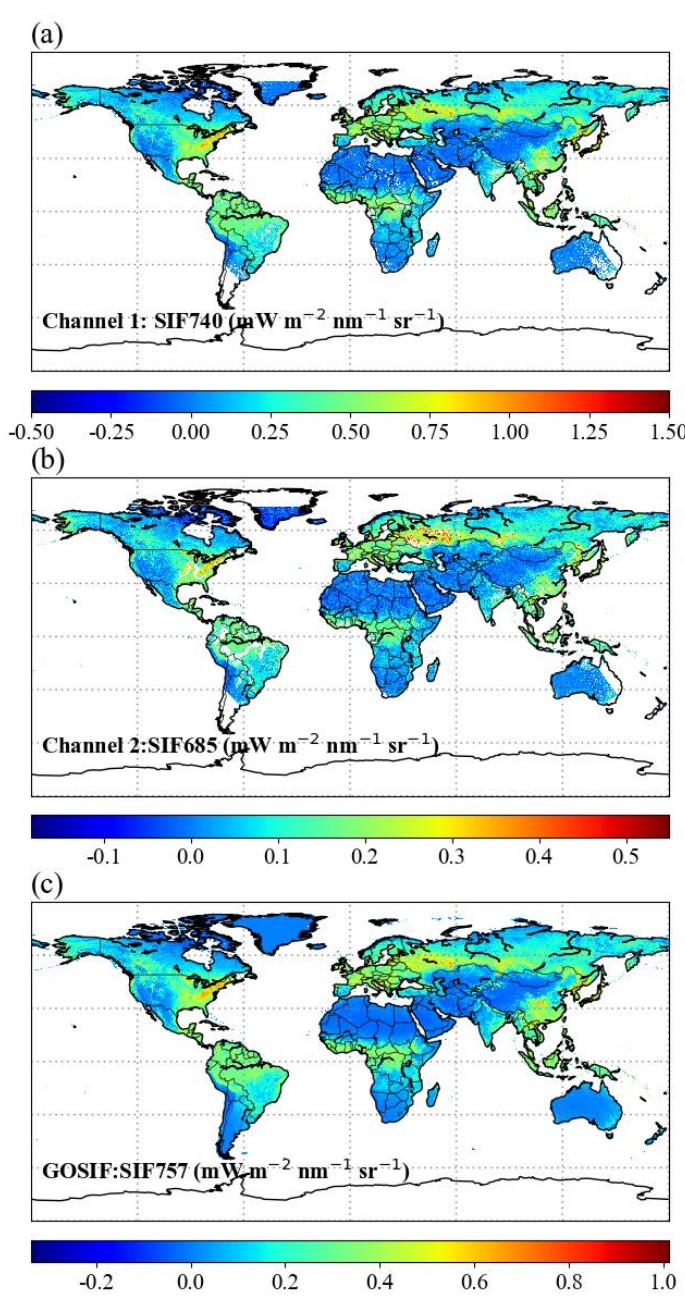

3    **Figure A1.** Global maps of mean SIF for each of the two channels, with GOSIF serving as the reference for true SIF. The SIF range is scaled

4    according to the proportion of the SIF spectrum used. The simulations represent 4-day time averages for June.



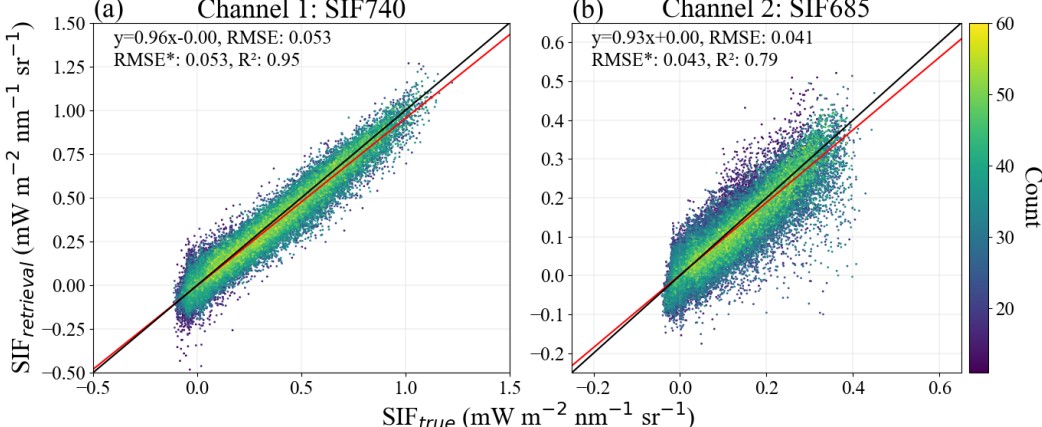

2 **Figure A2.** Global simulations of SIF retrieval from two channels, under the same conditions as Fig. A1. The black line denotes 'y = x', while the

3 red one denotes the linear fitting line.



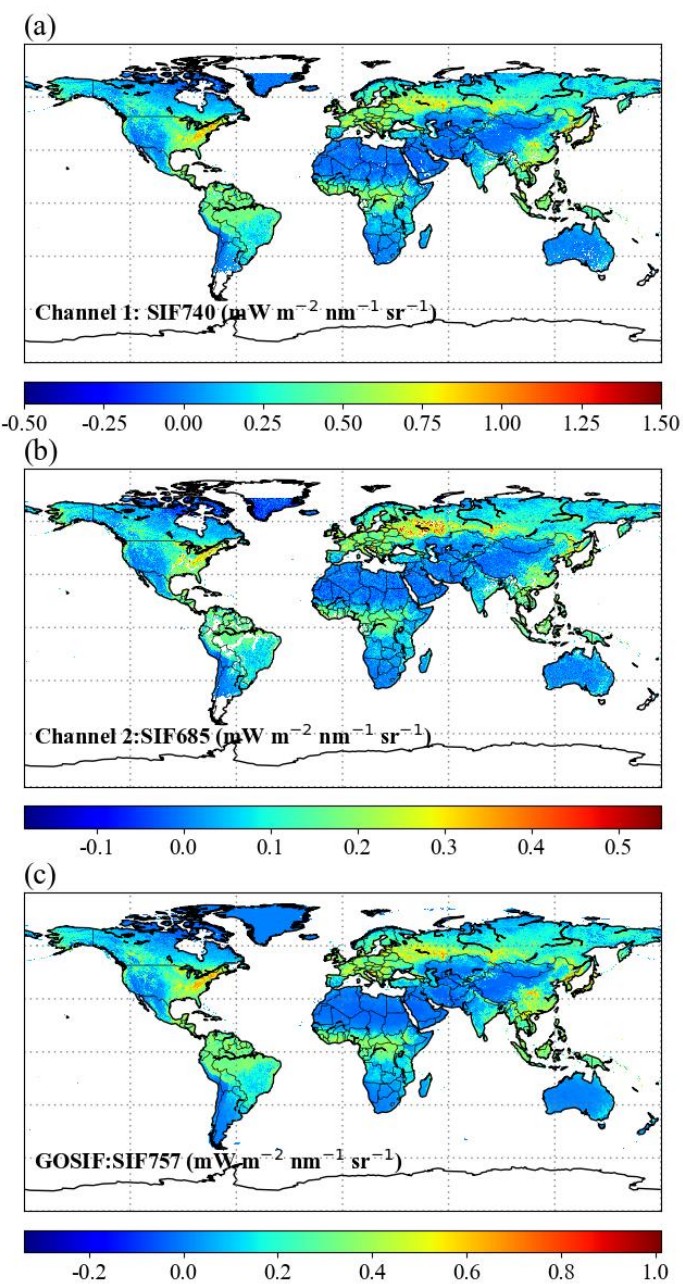

2   **Figure A3.** Global maps of mean SIF for each of the two channels, with GOSIF serving as the reference for true SIF. The SIF range is scaled

3   according to the proportion of the SIF spectrum utilized. The simulations represent 8-day averages for June.



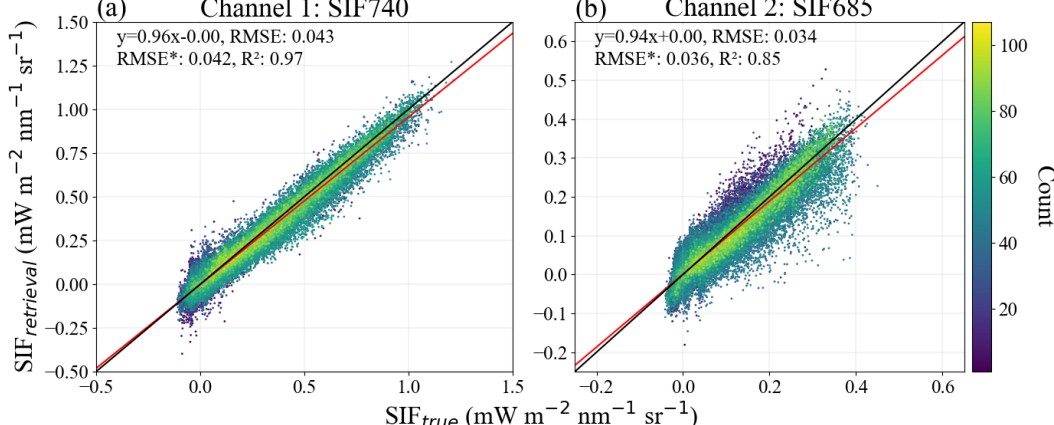

**Figure A4.** Global simulations of SIF retrieval from two channels, under the same conditions as Fig. A3. The black line denotes 'y = x', while the red one denotes the linear fitting line.

*Author contributions.* DZ, SD, and LL designed the experiments and DZ carried them out. DZ developed the model code and prepared the manuscript. CZ, MF, and YD contributed significantly to the research method and the manuscript revision. LT provided important support concerning the descriptions of the satellite.

*Competing interests.* The authors declare no conflicts of interest

*Financial support.* This work was funded by the National Key Research and Development Program Earth Observation and Navigation Key Project (grant no. 2023YFB3907405) and the National Natural Science Foundation of China (grant no. 42201356).



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
