# Peer review of "TanSat-2: a new satellite for mapping solar-induced chlorophyll fluorescence at both red and far-red bands with high spatio-temporal resolution"

_EGUsphere, 2024_

## Referee Comment (RC1)

Assessment of paper                                    March 12 2025

TanSat-2 A new satellite for accurate mapping solar induced chlorophyll fluorescence at both red and far-red bands with high spatial-temporal resolution

By Henry Buijs PhD.

Principal criteria

Scientific significance is ***excellent***

Scientific Quality is ***excellent***

Presentation quality ***Good***

The authors realize that Global observations of chlorophyll fluorescence (SIF) as first observed in ground reflected solar spectra measured by the Japanese GOSAT satellite can serve as a proxy for monitoring vegetation for photosynthetic activity as well as monitoring an significant part of the terrestrial carbon cycle.

The paper deals with modeling the measurement parameters in support of the development of a new satellite, TanSat-2, that permits to more accurately map chlorophyll fluorescence and thereby obtain a more accurate inventory of terrestrial vegetation and its effect on the Carbon cycle.

The paper is well organized.

Part 1 provides an introduction and background to the current status of realization in the subject field.

Part 2 Materials; provides the background to the work presented including the parameters of the planned TanSat-2 mission, simulation experiments and data, and an end to end orbit simulation dataset.

Part 4 Results; explains clearly that the analysis method is based on empirical data. It includes an independent validation with data not used in the modeling.

Part 5. Discussion.

The issue of cloud interference is of considerable importance. Especially over a wide swath as is planned for TanSat-2. The fraction of clear sky measurements gets to be quite small. Adding a cloud imaging camera could be beneficial to permit processing of identified cloud-free segments of each swath. The statement of not having incorporated rotational Raman scattering is probably not required since this occurs mainly at shorter wavelengths and is quite likely negligible in both regions of SIF. However, having made a statement about rotational Raman scattering, it is recommended that the authors make a cursory evaluation of its significance.

The first part of the paper deals with the derivation of a mathematical model that permits accurate computation of the intensity of fluorescence validated with a subset of available satellite data. The derivation follows well established mathematical methods such principal component analysis and is verified with additional satellite data. The model is used to guide the development of Tansat-2 including a planned elliptical orbit that, according to the authors, shall somewhat favor the more

populated Northern hemisphere. This is a problematic part of the paper. The highly elliptical orbit suggested for Tansat-2A does not appear to me an optimal choice. Whereas it will limit the global coverage to favor the Northern Hemisphere, and be Sun-synchronized around mid-day, it will seriously affect the uniformity of ground coverage. Near the apogee of the orbit, the swath size will be ten times larger than at perigee and the orbital motion will be significantly slower than at perigee.

I feel that the sun-synchronous elliptical orbit with an apogee approximately 10x higher than the perigee is not efficient and may lead to field of view aberrations that could compromise the accuracy of measurements. As well the swath width at apogee is much wider than at perigee making its ground coverage incomplete and difficult to fill out. I recommend that the authors describe in more detail the observational consequences of their choice of orbit. It seems to me that a near circular sun-synchronous orbit is more advantageous despite the overpass of more territory that is of less interest.

I recommend that the paper is important enough to be published.

Does the paper address relevant scientific questions within the scope of AMT? Yes

Does the paper present novel concepts, ideas, tools, or data? Yes

Are substantial conclusions reached? Yes

Are the scientific methods and assumptions valid and clearly outlined? Yes

Are the results sufficient to support the interpretations and conclusions? Some questions remain

Is the description of experiments and calculations sufficiently complete and precise to allow their reproduction by fellow scientists (traceability of results)? Yes

Do the authors give proper credit to related work and clearly indicate their own new/original contribution? Yes

Does the title clearly reflect the contents of the paper? Yes

Does the abstract provide a concise and complete summary? Yes

Is the overall presentation well structured and clear? Yes

Is the language fluent and precise? Yes

Are mathematical formulae, symbols, abbreviations, and units correctly defined and used? Yes

Should any parts of the paper (text, formulae, figures, tables) be clarified, reduced, combined, or eliminated? Yes see text above

Are the number and quality of references appropriate? Yes

Is the amount and quality of supplementary material appropriate? Yes

---

## Author Comment (AC1)

**Response to Comments from Reviewer 1**

Principal criteria

Scientific significance is excellent

Scientific Quality is excellent

Presentation quality Good

The authors realize that Global observations of chlorophyll fluorescence (SIF) as first observed in ground reflected solar spectra measured by the Japanese GOSAT satellite can serve as a proxy for monitoring vegetation for photosynthetic activity as well as monitoring an significant part of the terrestrial carbon cycle.

The paper deals with modeling the measurement parameters in support of the development of a new satellite, TanSat-2, that permits to more accurately map chlorophyll fluorescence and thereby obtain a more accurate inventory of terrestrial vegetation and its effect on the Carbon cycle.

The paper is well organized.

Part 1 provides an introduction and background to the current status of realization in the subject field.

Part 2 Materials; provides the background to the work presented including the parameters of the planned TanSat-2 mission, simulation experiments and data, and an end to end orbit simulation dataset.

Part 4 Results; explains clearly that the analysis method is based on empirical data. It includes an independent validation with data not used in the modeling.

Part 5. Discussion.

Thanks a lot for your encouraging words and helpful comments. We have carefully revised the manuscript according to your comments and suggestions, especially on the discussion of the limitations of our simulation framework and the shortcomings of the elliptical orbit for TanSat-2. The responses are in blue font, and the relevant revised parts of the manuscript are attached in purple font.

Comments 1. The issue of cloud interference is of considerable importance. Especially over a wide swath as is planned for TanSat-2. The fraction of clear sky measurements gets to be quite small. Adding a cloud imaging camera could be beneficial to permit processing of identified cloud-free segments of each swath. The statement of not having incorporated rotational Raman scattering is probably not required since this occurs mainly at shorter wavelengths and is quite likely negligible in both regions of SIF. However, having made a statement about rotational Raman scattering, it is recommended that the authors make a cursory evaluation of its significance.

Thank you for your valuable comments. The issue of cloud interference is indeed of significant importance. Cloud presence can substantially affect radiative transfer processes and, consequently, SIF retrievals. The CAPHI instrument onboard TanSat-2 will provide AOD and cloud coverage information, which will enhance our understanding of atmospheric conditions and support future SIF retrievals with TanSat-2.

Regarding radiative transfer modeling limitations, our simulations explicitly exclude rotational Raman scattering (RRS) due to the coupled complexities arising from its interdependencies with Mie scattering and nonlinear interactions with other atmospheric parameters. This omission arises from algorithmic constraints in the MODTRAN-based radiative transfer framework, which struggles to resolve such multi-scale scattering synergies. The RRS effects will be relatively small in the spectral range of red and far-red bans (Vasilkov et al., 2013). In data-driven SIF retrieval frameworks, its influence can be modeled. Consequently, it is incorporated into the basis vectors (Joiner et al., 2016). This inherent limitation of our simulation framework was analyzed in the Section 5.1.

In Section 5.1:

**It should be noted that our simulations did not account for radiative effects induced by RRS. The RRS intensity typically decreases with increasing wavelength (Vasilkov et al., 2013), and its spectral interference is statistically negligible within the red to far-red spectral bands. Moreover, the data-driven SIF retrieval framework inherently addresses potential RRS contamination through basis vector parameterization (Joiner et al., 2016). This approach enables the decoupling of RRS-induced spectral variations from SIF emission signals.**

Comments 2. The first part of the paper deals with the derivation of a mathematical model that permits accurate computation of the intensity of fluorescence validated with a subset of available satellite data. The derivation follows well established mathematical methods such principal component analysis and is verified with additional satellite data. The model is used to guide the development of Tansat-2 including a planned elliptical orbit that, according to the authors, shall somewhat favor the more populated Northern hemisphere. This is a problematic part of the paper. The highly elliptical orbit suggested for Tansat-2A does not appear to me an optimal choice. Whereas it will limit the global coverage to favor the Northern Hemisphere, and be Sun-synchronized around mid-day, it will seriously affect the uniformity of ground coverage. Near the apogee of the orbit, the swath size will be ten times larger than at perigee and the orbital motion will be significantly slower than at perigee.

I feel that the sun-synchronous elliptical orbit with an apogee approximately 10x higher than the perigee is not efficient and may lead to field of view aberrations that could compromise the accuracy of measurements. As well the swath width at apogee is much wider than at perigee making its ground coverage incomplete and difficult to fill out. I recommend that the authors describe in more detail the observational consequences of their choice of orbit. It seems to me that a near circular sunsynchronous orbit is more advantageous despite the overpass of more territory that is of less interest.

Thank you for your valuable comments. We truly appreciate your thorough analysis and well-considered concerns regarding the use of an elliptical orbit for TanSat-2. Your insights are highly relevant, and we totally agree your comments on the elliptical orbit, and add a paragraph on its shortcomings in the Section 5.1. Furthermore, we would like to add some details to clarify the scientific rationale behind our orbital choice in the response letter.

TanSat-2 is designed to facilitate global carbon stocktaking. The choice of the inclined elliptical orbit is intended to ensure more frequent coverage of the densely populated Northern Hemisphere, particularly key regions such as Asia, North America, and Europe. These regions are of paramount scientific importance for global carbon inventory assessments, especially in the monitoring of atmospheric gases like carbon dioxide. The satellite's trajectory is optimized to ensure that the Northern Hemisphere benefits from more frequent and consistent data collection.

However, as you pointed out, the use of such an elliptical orbit may present challenges. The efficiency of the sun-synchronous elliptical orbit is relatively low, as the apogee is approximately 10 times higher than the perigee. Although the imaging setting only allows observation for orbital altitudes above ~2,350 km, significant altitude disparity still persists. This results in a swath width variation exceeding twofold, causing uneven ground coverage and reduced efficiency in achieving uniform global sampling—particularly pronounced in equatorial zones. Additionally, this orbit may lead to field of view aberrations, which can impact the accuracy of measurements. These shortcomings must be carefully considered during the orbit design and satellite system design phases.

In Section 5.1:

**For the end-to-end orbit simulation, we modeled TanSat-2's Earth observations based on its orbital parameters. Designed to facilitate global carbon stocktaking, the inclined elliptical orbit enhances observational frequency over the Northern Hemisphere's densely populated regions (e.g., Asia, North America, and Europe), as evidenced by the increased observation density shown in Figures 11 and A1. Continuous observations over four- or eight-day cycles ensure near-global coverage. However, this orbital architecture introduces inherent challenges. The efficiency of the sun-synchronous elliptical orbit is relatively low, as its apogee is approximately ten times higher than its perigee. Although the imaging setting only allow observation for orbital altitudes above ~2,350 km, significant altitude disparity still persists. This results in a swath width variation exceeding twofold, leading to uneven ground coverage and spatial resolution, which in turn reduces the efficiency of achieving uniform global sampling—particularly in equatorial regions. Furthermore, the orbit may induce field of view aberrations that could compromise measurement accuracy. These limitations necessitate systematic mitigation strategies during satellite system design and orbital parameter optimization.**

---

## Author Comment (AC2)

**Response to Comments from Reviewer 2**

This study presents an evaluation of the TanSat-2 satellite's capabilities for dual-band solar-induced chlorophyll fluorescence (SIF) retrieval through spectral simulations and end-to-end orbit simulations. The research demonstrates a well-structured approach, rigorous methodological design, and scientifically valuable findings. It is recommended for acceptance after addressing the following revisions:

Thanks a lot for your encouraging words and helpful comments. We have carefully revised the manuscript according to your comments and suggestions, especially on the analysis of the potential biases or regional variations in retrieval errors and the descriptions of the data preprocessing steps. The responses are in blue font, and the relevant revised parts of the manuscript are attached in purple font.

Comments 1. Line numbers should be consistent in the whole manuscript not start with 1 for each page.

Thank you for bringing to our attention the need for consistent line numbering. We have applied uniform line numbering throughout the entire manuscript to ensure clarity and ease of reference. We appreciate your feedback and hope that this adjustment enhances readability.

Comments 2. For section 4.2: The manuscript lacks detailed descriptions of data preprocessing steps, particularly in the end-to-end orbit simulations. For example, cloud contamination were not explicitly modeled, and the impact of cloud screening on SIF retrieval accuracy remains unclear. It is recommended to supplement details on cloud screening strategies and their implications for retrieval performance.

Thank you for your valuable comments. We have provided a more detailed description of the data preprocessing steps in the simulation experiments section, specifically in Sections 2.2.1 and 2.2.3. The issue of cloud interference is indeed of significant importance. Cloud presence can substantially affect radiative transfer processes and, consequently, SIF retrievals. In our study, we did not explicitly model cloud contamination or use cloud cover products for screening. The atmospheric radiative transfer framework based on MODTRAN is not well-suited for accurately simulating the effects of cloud contamination. Our primary objective is to highlight the advantages of the TanSat-2 sensor's enhanced spectral capabilities and its high temporal and spatial resolution for SIF retrieval. As such, our end-to-end orbital simulation assumes clear-sky conditions without incorporating cloud effects. Furthermore, the CAPHI instrument onboard TanSat-2 will provide aerosol optical depth (AOD) and cloud coverage information, which will enhance our understanding of atmospheric conditions and support future SIF retrievals with TanSat-2. These considerations are also described in Section 2.2.3.

In Section 2.2.1:

**The Soil Canopy Observation Photochemistry and Energy Flux (SCOPE) model (van der Tol et al., 2009) is capable of simulating vegetation canopy reflectance spectra and SIF under diverse canopy structures and leaf biochemical conditions,**

including leaf optical properties (e.g., chlorophyll content, dry matter) and canopy structural parameters (e.g., leaf area index, canopy height). Atmospheric radiative transfer functions were derived from the Moderate-resolution Atmospheric TRANsmission model (MODTRAN5; Berk et al., 1998, 2000) to generate TOA radiance. Critical processing steps included: (1) using the MODTRAN Interrogation Technique (MIT) (Verhoef and Bach, 2012; Verhoef et al., 2018) to extract 18 spectral transfer functions parameterized by aerosol optical depth, water vapor content, and observation geometry from MODTRAN5 outputs; (2) using the RTMo module in SCOPE, which dynamically couples these functions with bidirectional reflectance distribution (BRDF) and SIF emission spectra to resolve surface-atmosphere interactions. The simulated atmospheric parameters, canopy reflectance, and SIF signals were then integrated using the radiative transfer operator defined in Equation 1 to generate the TOA radiance spectra across the 640–850 nm range.

In Section 2.2.3:

It should be noted that we did not explicitly simulate cloud contamination or utilize cloud cover products for screening purposes. As such, our end-to-end orbital simulation operates under the assumption of clear sky conditions, deliberately excluding the potential impacts of cloud cover. Furthermore, the CAPHI instrument aboard the satellite will supply data on AOD and cloud coverage, enhancing our understanding of atmospheric conditions and cloud dynamics, while also providing supplementary data for future SIF retrieval efforts with TanSat-2. Furthermore, our simulations exclude rotational Raman scattering (RRS) effects. The RRS effects will be relatively small in the spectral range of red and far-red band (Vasilkov et al., 2013). Fig. 4 presents pseudo-color composites using the near-infrared, red, and green bands of MCD43C4, comparing surface reflectance reconstructions with simulated TOA radiances in seven geomorphologically distinct regions, including desert, boreal forest, and tropical rainforest ecosystems.

Comments 3. For page 16: The manuscript mentions that differences between retrieved SIF and GOSIF inputs were generally within 0.15 mW m$^{-2}$ sr$^{-1}$ nm$^{-1}$, yet it does not explore potential biases or regional variations in retrieval errors. A more detailed discussion on possible sources of discrepancies (e.g., land cover types, atmospheric effects) would strengthen the validity of the results.

Thank you for your comments. We have provided a more detailed description in the current version of the manuscript regarding potential biases in SIF retrieval errors, their regional variations, and possible sources of these errors (Section 4.2).

In Section 4.2:

The retrieval uncertainties manifest predominantly as spectrally structured noise coupled with pronounced sensitivity to atmospheric scattering processes. The former primarily varies with scene-specific radiance magnitudes within retrieval windows, while systematic SIF underestimation in high aerosol-loading

**regimes arises from unaccounted scattering effects within our forward model. In order to disentangle the error sources, we conducted a statistical analysis of the far-red SIF retrieval errors (results for the red band are similar) in relation to AOD, the albedo within the fitting window, SZA, and VZA, as shown in Fig. 15. The results indicate that RMSE increases significantly with the increase in surface albedo and the decrease in SZA. Both changes enhance the background radiance, leading to a noticeable rise in retrieval uncertainty due to an increase in signal noise. Spatially, these retrieval errors dominate in bright surface areas, such as the Sahara Desert and the Congo Basin, as shown in Fig. 13. Meanwhile, retrieval bias exhibits substantial amplification under higher AOD and larger VZA. This phenomenon arises because the higher AOD strengthens atmospheric scattering efficiency, while larger VZA values extend the effective radiative path length. These factors collectively amplify atmospheric scattering effects, resulting in progressively larger underestimation of SIF. The spatial pattern of this bias prominently features regions with high aerosol, particularly Central and South Asia, as depicted in Fig. 13. Notably, red-SIF retrieval over aquatic environments necessitates distinct processing techniques compared to terrestrial environments. This technical disparity manifests as substantially exaggerated SIF estimates within specific watersheds, particularly those in northern/western Russia and the Great Lakes region of North America.**

[Figure]

**Figure 15. Statistical analysis of far-red SIF retrieval errors with respect to (a) AOD, (b) far-red band albedo, (c) SZA, and (d) VZA. The bias represents the mean value of ΔSIF (retrieved SIF - true SIF).**

Comments 4. For appendix Figure A2 & A4: The mentioned figures showing 4-day and 8-day composites. They are mentioned in the results but are not adequately described in the main text. It is recommended to integrate a brief discussion of their significance and ensure that all supplementary figures are clearly referenced.

Thank you for your comments. The results of the 4-day and 8-day composites are described in more detail in Section 4.2 of the main text. Figures A1 to A4 have been combined into two figures, with clear references to them.

In Section 4.2:

Moreover, we evaluated the results of 4-day and 8-day composites, as shown in Fig A1 and A2. Compared to single-day observations, the global coverage is more comprehensive, and the composite method significantly reduces retrieval errors by increasing observation density and suppressing noise (the RMSE for the 4-day and 8-day composites decreased by 35% and 47%, respectively). This further demonstrates the performance of the instrument on the TanSat-2 satellite. However, several persistent systematic biases were detected: underestimation in high AOD regions (Central Asia/South Asia) due to enhanced atmospheric scattering, overestimation in bright surfaces (Sahara Desert, Congo Basin) caused by radiation saturation, and overestimation of red SIF near water bodies (Russian rivers, Great Lakes in North America) due to residual surface reflectance effects. Overall, the 4-day and 8-day composites achieved excellent accuracy, demonstrating TanSat-2 robust retrieval performance that balances spatial coverage and accuracy. The $R^2$ values for the two channels were 0.95 and 0.79, and 0.97 and 0.85, respectively, while the RMSE values were 0.053 and 0.041, and 0.043 and 0.034 mW m$^{-2}$ sr$^{-1}$ nm$^{-1}$. It should be noted that although the AOD product used partially accounts for cloud impacts, it does not explicitly model cloud contamination or apply cloud fraction products for screening. Consequently, our results assume clear-sky conditions. Additionally, these simulations do not incorporate RRS, as the filter only considers data with a SZA less than 70°, where the RRS effects are minimal.

[Figure]

Figure A1. Global maps of TanSat-2 observation counts (top panels) and retrieval

**errors (retrieved SIF minus reference SIF) for Channel l (middle panels) and Channel 2 (bottom panels). Results derive from 4-day (left column) and 8-day (right column) composites at 0.05° grid resolution.**

[Figure]

**Figure A2. Retrieved versus reference SIF relationships for TanSat-2's two channels derived from 4-day composites (a, b) and 8-day composites (c, d). Black lines denote the 1:1 relationship, while red lines indicate linear regression fits.**

Comments 5. While the manuscript is well-structured, some sections contain lengthy technical descriptions that could benefit from clearer segmentation. Additionally, minor grammatical inconsistencies exist, particularly in the descriptions of retrieval equations. Comprehensive language polishing is advised to improve readability and ensure consistency in technical terminology.

Thank you for your comments. We have reorganized the lengthy technical descriptions into more coherent sections and performed a thorough language revision throughout the manuscript. The following are examples of our revisions in Sections 2.2.3 and 3.1.

In Section 2.2.3:

Before revisions:

[revised manuscript text omitted]